# Effective Parallelisation for Machine Learning

**Michael Kamp**
University of Bonn
and Fraunhofer IAIS
kamp@cs.uni-bonn.de

**Mario Boley**
Max Planck Institute for Informatics
and Saarland University
mboley@mpi-inf.mpg.de

**Olana Missura**
Google Inc.
olanam@google.com

**Thomas Gärtner**
University of Nottingham
thomas.gaertner@nottingham.ac.uk

## Abstract

We present a novel parallelisation scheme that simplifies the adaptation of learning algorithms to growing amounts of data as well as growing needs for accurate and confident predictions in critical applications. In contrast to other parallelisation techniques, it can be applied to a broad class of learning algorithms without further mathematical derivations and without writing dedicated code, while at the same time maintaining theoretical performance guarantees. Moreover, our parallelisation scheme is able to reduce the runtime of many learning algorithms to polylogarithmic time on quasi-polynomially many processing units. This is a significant step towards a general answer to an open question on the efficient parallelisation of machine learning algorithms in the sense of Nick's Class (NC). The cost of this parallelisation is in the form of a larger sample complexity. Our empirical study confirms the potential of our parallelisation scheme with fixed numbers of processors and instances in realistic application scenarios.

## 1 Introduction

This paper contributes a novel and provably effective parallelisation scheme for a broad class of learning algorithms. The significance of this result is to allow the confident application of machine learning algorithms with growing amounts of data. In critical application scenarios, i.e., when errors have almost prohibitively high cost, this confidence is essential [27, 36]. To this end, we consider the parallelisation of an algorithm to be effective if it achieves the same confidence and error bounds as the sequential execution of that algorithm in much shorter time. Indeed, our parallelisation scheme can reduce the runtime of learning algorithms from polynomial to polylogarithmic. For that, it consumes more data and is executed on a quasi-polynomial number of processing units.

To formally describe and analyse our parallelisation scheme, we consider the regularised risk minimisation setting. For a fixed but unknown joint probability distribution $\mathcal{D}$ over an *input space* $\mathcal{X}$ and an *output space* $\mathcal{Y}$, a dataset $D \subseteq \mathcal{X} \times \mathcal{Y}$ of size $N \in \mathbb{N}$ drawn iid from $\mathcal{D}$, a convex *hypothesis space* $\mathcal{F}$ of functions $f \colon \mathcal{X} \to \mathcal{Y}$, a loss function $\ell \colon \mathcal{F} \times \mathcal{X} \times \mathcal{Y} \to \mathbb{R}$ that is convex in $\mathcal{F}$, and a convex regularisation term $\Omega \colon \mathcal{F} \to \mathbb{R}$, *regularised risk minimisation algorithms* solve

$$\mathcal{L}(D) = \operatorname*{argmin}_{f \in \mathcal{F}} \sum_{(x,y) \in D} \ell(f, x, y) + \Omega(f) \ . \tag{1}$$

The aim of this approach is to obtain a hypothesis $f \in \mathcal{F}$ with small *regret*

$$\mathcal{Q}(f) = \mathbb{E}\left[\ell(f, x, y)\right] - \operatorname*{argmin}_{f' \in \mathcal{F}} \mathbb{E}\left[\ell(f', x, y)\right] \ . \tag{2}$$

Regularised risk minimisation algorithms are typically designed to be *consistent* and *efficient*. They are consistent if there is a function $N_0 \colon \mathbb{R}_+ \times \mathbb{R}_+ \to \mathbb{R}_+$ such that for all $\varepsilon > 0$, $\Delta \in (0, 1]$, $N \in \mathbb{N}$ with $N \geq N_0(\varepsilon, \Delta)$, and training data $D \sim \mathcal{D}^N$, the probability of generating an $\varepsilon$-bad hypothesis is no greater than $\Delta$, i.e.,

$$P\left(\mathcal{Q}\left(\mathcal{L}(D)\right) > \varepsilon\right) \leq \Delta \ . \tag{3}$$

They are efficient if the *sample complexity* $N_0(\varepsilon, \Delta)$ is polynomial in $1/\varepsilon$, $\log 1/\Delta$ and the *runtime complexity* $T_{\mathcal{L}}$ is polynomial in the sample complexity. This paper considers the parallelisation of such consistent and efficient learning algorithms, e.g., support vector machines, regularised least squares regression, and logistic regression. We additionally assume that data is abundant and that $\mathcal{F}$ can be parametrised in a fixed, finite dimensional Euclidean space $\mathbb{R}^d$ such that the convexity of the regularised risk minimisation problem (Equation 1) is preserved. In other cases, (non-linear) low-dimensional embeddings [2, 28] can preprocess the data to facilitate parallel learning with our scheme. With slight abuse of notation, we identify the hypothesis space with its parametrisation.

The main theoretical contribution of this paper is to show that algorithms satisfying the above conditions can be parallelised *effectively*. We consider a parallelisation to be effective if the $(\varepsilon, \Delta)$-guarantees (Equation 3) are achieved in time polylogarithmic in $N_0(\varepsilon, \Delta)$. The cost for achieving this reduction in runtime comes in the form of an increased data size and in the number of processing units used. For the parallelisation scheme presented in this paper, we are able to bound this cost by a quasi-polynomial in $1/\varepsilon$ and $\log 1/\Delta$. The main practical contribution of this paper is an effective parallelisation scheme that treats the underlying learning algorithm as a *black-box*, i.e., it can be parallelised without further mathematical derivations and without writing dedicated code.

Similar to averaging-based parallelisations [32, 45, 46], we apply the underlying learning algorithm in parallel to random subsets of the data. Each resulting hypothesis is assigned to a leaf of an aggregation tree which is then traversed bottom-up. Each inner node computes a new hypothesis that is a *Radon point* [30] of its children's hypotheses. In contrast to aggregation by averaging, the Radon point increases the confidence in the aggregate doubly-exponentially with the height of the aggregation tree. We describe our parallelisation scheme, the *Radon machine*, in detail in Section 2. Comparing the Radon machine to the underlying learning algorithm which is applied to a dataset of the size necessary to achieve the same confidence, we are able to show a reduction in runtime from polynomial to polylogarithmic in Section 3.

The empirical evaluation of the Radon machine in Section 4 confirms its potential in practical settings. Given the same amount of data as the underlying learning algorithm, the Radon machine achieves a substantial reduction of computation time in realistic applications. Using 150 processors, the Radon machine is between 80 and around 700-times faster than the underlying learning algorithm on a single processing unit. Compared with parallel learning algorithms from Spark's MLlib, it achieves hypotheses of similar quality, while requiring only $15 - 85\%$ of their runtime.

Parallel computing [18] and its limitations [13] have been studied for a long time in theoretical computer science [7]. Parallelising polynomial time algorithms ranges from being 'embarrassingly' [26] easy to being believed to be impossible. For the class of decision problems that are the hardest in P, i.e., for P-complete problems, it is believed that there is no efficient parallel algorithm in the sense of Nick's Class (NC [9]): efficient parallel algorithms in this sense are those that can be executed in *polylogarithmic time* on a *polynomial number of processing units*. Our paper thus contributes to

---

**Algorithm 1** Radon Machine

---

**Input:** learning algorithm $\mathcal{L}$, dataset $D \subseteq \mathcal{X} \times \mathcal{Y}$, Radon number $r \in \mathbb{N}$, and parameter $h \in \mathbb{N}$
**Output:** hypothesis $f \in \mathcal{F}$
  1: **divide** $D$ into $r^h$ iid subsets $D_i$ of roughly equal size
  2: **run** $\mathcal{L}$ in parallel to obtain $f_i = \mathcal{L}(D_i)$
  3: $S \leftarrow \{f_1, \ldots, f_{r^h}\}$
  4: **for** $i = h - 1, \ldots, 1$ **do**
  5:     **partition** $S$ into iid subsets $S_1, \ldots, S_{r^i}$ of size $r$ each
  6:     **calculate** Radon points $\mathfrak{r}(S_1), \ldots, \mathfrak{r}(S_{r^i})$ in parallel    *# see Definition 1 and Appendix C.1*
  7:     $S \leftarrow \{\mathfrak{r}(S_1), \ldots, \mathfrak{r}(S_{r^i})\}$
  8: **end for**
  9: **return** $\mathfrak{r}(S)$

---

understanding the extent to which efficient parallelisation of polynomial time learning algorithms is possible. This connection and other approaches to parallel learning are discussed in Section 5.

## 2    From Radon Points to Radon Machines

The Radon machine, described in Algorithm 1, first executes the underlying (base) learning algorithm on random subsets of the data to quickly achieve weak hypotheses and then iteratively aggregates them to stronger ones. Both the generation of weak hypotheses and the aggregation can be executed in parallel. To aggregate hypotheses, we follow along the lines of the iterated Radon point algorithm which was originally devised to approximate the centre point, i.e., a point of largest Tukey depth [38], of a finite set of points [8]. The Radon point [30] of a set of points is defined as follows:

**Definition 1.** *A* Radon partition *of a set* $S \subset \mathcal{F}$ *is a pair* $A, B \subset S$ *such that* $A \cap B = \emptyset$ *but* $\langle A \rangle \cap \langle B \rangle \neq \emptyset$, *where* $\langle \cdot \rangle$ *denotes the convex hull. The* Radon number *of a space* $\mathcal{F}$ *is the smallest* $r \in \mathbb{N}$ *such that for all* $S \subset \mathcal{F}$ *with* $|S| \geq r$ *there is a Radon partition; or* $\infty$ *if no* $S \subset \mathcal{F}$ *with Radon partition exists. A* Radon point *of a set* $S$ *with Radon partition* $A, B$ *is any* $\mathfrak{r} \in \langle A \rangle \cap \langle B \rangle$.

We now present the Radon machine (Algorithm 1), which is able to effectively parallelise consistent and efficient learning algorithms. Input to this parallelisation scheme is a learning algorithm $\mathcal{L}$ on a hypothesis space $\mathcal{F}$, a dataset $D \subseteq \mathcal{X} \times \mathcal{Y}$, the Radon number $r \in \mathbb{N}$ of the hypothesis space $\mathcal{F}$, and a parameter $h \in \mathbb{N}$. It divides the dataset into $r^h$ subsets $D_1, \dots, D_{r^h}$ (line 1) and runs the algorithm $\mathcal{L}$ on each subset in parallel (line 2). Then, the set of hypotheses (line 3) is iteratively aggregated to form better sets of hypotheses (line 4-8). For that the set is partitioned into subsets of size $r$ (line 5) and the Radon point of each subset is calculated in parallel (line 6). The final step of each iteration is to replace the set of hypotheses by the set of Radon points (line 7).

The scheme requires a hypothesis space with a valid notion of convexity and finite Radon number. While other notions of convexity are possible [16, 33], in this paper we restrict our consideration to Euclidean spaces with the usual notion of convexity. Radon's theorem [30] states that the Euclidean space $\mathbb{R}^d$ has Radon number $r = d + 2$. Radon points can then be obtained by solving a system of linear equations of size $r \times r$ (to be fully self-contained we state the system of linear equations explicitly in Appendix C.1). The next proposition gives a guarantee on the quality of Radon points:

**Proposition 2.** *Given a probability measure* $P$ *over a hypothesis space* $\mathcal{F}$ *with finite Radon number* $r$, *let* $F$ *denote a random variable with distribution* $P$. *Furthermore, let* $\mathfrak{r}$ *be the random variable obtained by computing the Radon point of* $r$ *random points drawn according to* $P^r$. *Then it holds for the expected regret* $\mathcal{Q}$ *and all* $\varepsilon \in \mathbb{R}$ *that*

$$P\left(\mathcal{Q}\left(\mathfrak{r}\right) > \varepsilon\right) \leq \left(r P\left(\mathcal{Q}\left(F\right) > \varepsilon\right)\right)^2 \quad .$$

The proof of Proposition 2 is provided in Section 7. Note that this proof also shows the robustness of the Radon point compared to the average: if only one of $r$ points is $\varepsilon$-bad, the Radon point is still $\varepsilon$-good, while the average may or may not be; indeed, in a linear space with any set of $\varepsilon$-good hypotheses and any $\varepsilon' \geq \varepsilon$, we can always find a single $\varepsilon'$-bad hypothesis such that the average of all these hypotheses is $\varepsilon'$-bad.

A direct consequence of Proposition 2 is a bound on the probability that the output of the Radon machine with parameter $h$ is bad:

**Theorem 3.** *Given a probability measure* $P$ *over a hypothesis space* $\mathcal{F}$ *with finite Radon number* $r$, *let* $F$ *denote a random variable with distribution* $P$. *Denote by* $\mathfrak{r}_1$ *the random variable obtained by computing the Radon point of* $r$ *random points drawn iid according to* $P$ *and by* $P_1$ *its distribution. For any* $h \in \mathbb{N}$, *let* $\mathfrak{r}_h$ *denote the Radon point of* $r$ *random points drawn iid from* $P_{h-1}$ *and by* $P_h$ *its distribution. Then for any convex function* $\mathcal{Q} : \mathcal{F} \to \mathbb{R}$ *and all* $\varepsilon \in \mathbb{R}$ *it holds that*

$$P\left(\mathcal{Q}(\mathfrak{r}_h) > \varepsilon\right) \leq \left(r P\left(\mathcal{Q}(F) > \varepsilon\right)\right)^{2^h} \quad .$$

The proof of Theorem 3 is also provided in Section 7. For the Radon machine with parameter $h$, Theorem 3 shows that the probability of obtaining an $\varepsilon$-bad hypothesis is doubly exponentially reduced: with a bound $\delta$ on this probability for the base learning algorithm, the bound $\Delta$ on this probability for the Radon machine is

$$\Delta = \left(r\delta\right)^{2^h} \quad . \tag{4}$$

In the next section we compare the Radon machine to its base learning algorithm which is applied to a dataset of the size necessary to achieve the same $\varepsilon$ and $\Delta$.

## 3 Sample and Runtime Complexity

In this section we first derive the sample and runtime complexity of the Radon machine $\mathcal{R}$ from the sample and runtime complexity of the base learning algorithm $\mathcal{L}$. We then relate the runtime complexity of the Radon machine to an application of the base learning algorithm which achieves the same $(\varepsilon, \Delta)$-guarantee. For that, we consider consistent and efficient base learning algorithms with a sample complexity of the form $N_0^{\mathcal{L}}(\varepsilon, \delta) = (\alpha_\varepsilon + \beta_\varepsilon \operatorname{ld} 1/\delta)^k$, for some[1] $\alpha_\varepsilon, \beta_\varepsilon \in \mathbb{R}$, and $k \in \mathbb{N}$. From now on, we also assume that $\delta \leq 1/2r$ for the base learning algorithm.

The Radon machine creates $r^h$ base hypotheses and, with $\Delta$ as in Equation 4, has sample complexity

$$N_0^{\mathcal{R}}(\varepsilon, \Delta) = r^h N_0^{\mathcal{L}}(\varepsilon, \delta) = r^h \cdot \left( \alpha_\varepsilon + \beta_\varepsilon \operatorname{ld} \frac{1}{\delta} \right)^k \quad . \tag{5}$$

Theorem 3 then implies that the Radon machine with base learning algorithm $\mathcal{L}$ is consistent: with $N \geq N_0^{\mathcal{R}}(\varepsilon, \Delta)$ samples it achieves an $(\varepsilon, \Delta)$-guarantee.

To achieve the same guarantee as the Radon machine, the application of the base learning algorithm $\mathcal{L}$ itself (sequentially) would require $M \geq N_0^{\mathcal{L}}(\varepsilon, \Delta)$ samples, where

$$N_0^{\mathcal{L}}(\varepsilon, \Delta) = N_0^{\mathcal{L}}\left( \varepsilon, (r\delta)^{2^h} \right) = \left( \alpha_\varepsilon + 2^h \cdot \beta_\varepsilon \operatorname{ld} \frac{1}{r\delta} \right)^k \quad . \tag{6}$$

For base learning algorithms $\mathcal{L}$ with runtime $T_{\mathcal{L}}(n)$ polynomial in the data size $n \in \mathbb{N}$, i.e., $T_{\mathcal{L}}(n) \in \mathcal{O}(n^\kappa)$ with $\kappa \in \mathbb{N}$, we now determine the runtime $T_{\mathcal{R},h}(N)$ of the Radon machine with $h$ iterations and $c = r^h$ processing units on $N \in \mathbb{N}$ samples. In this case all base learning algorithms can be executed in parallel. In practical applications fewer physical processors can be used to simulate $r^h$ processing units—we discuss this case in Section 5.

The runtime of the Radon machine can be decomposed into the runtime of the base learning algorithm and the runtime for the aggregation. The base learning algorithm requires $n \geq N_0^{\mathcal{L}}(\varepsilon, \delta)$ samples and can be executed on $r^h$ processors in parallel in time $T_{\mathcal{L}}(n)$. The Radon point in each of the $h$ iterations can then be calculated in parallel in time $r^3$ (see Appendix C.1). Thus, the runtime of the Radon machine with $N = r^h n$ samples is

$$T_{\mathcal{R},h}(N) = T_{\mathcal{L}}(n) + hr^3 \quad . \tag{7}$$

In contrast, the runtime of the base learning algorithm for achieving the same guarantee is $T_{\mathcal{L}}(M)$ with $M \geq N_0^{\mathcal{L}}(\varepsilon, \Delta)$. Ignoring logarithmic and constant terms, $N_0^{\mathcal{L}}(\varepsilon, \Delta)$ behaves as $2^h N_0^{\mathcal{L}}(\varepsilon, \delta)$. To obtain polylogarithmic runtime of $\mathcal{R}$ compared to $T_{\mathcal{L}}(M)$, we choose the parameter $h \approx \operatorname{ld} M - \operatorname{ld} \operatorname{ld} M$ such that $n \approx M/2^h = \operatorname{ld} M$. Thus, the runtime of the Radon machine is in $\mathcal{O}\left( \operatorname{ld}^\kappa M + r^3 \operatorname{ld} M \right)$. This result is formally summarised in Theorem 4.

**Theorem 4.** *The Radon machine with a consistent and efficient regularised risk minimisation algorithm on a hypothesis space with finite Radon number has polylogarithmic runtime on quasi-polynomially many processing units if the Radon number can be upper bounded by a function poly-logarithmic in the sample complexity of the efficient regularised risk minimisation algorithm.*

The theorem is proven in Appendix A.1 and relates to Nick's Class [1]: A decision problem can be solved efficiently in parallel in the sense of Nick's Class, if it can be decided by an algorithm in polylogarithmic time on polynomially many processors (assuming, e.g., PRAM model). For the class of decision problems that are the hardest in $P$, i.e., for $P$-complete problems, it is believed that there is no efficient parallel algorithm for solving them in this sense. Theorem 4 provides a step towards finding efficient parallelisations of regularised risk minimisers and towards answering the open question: is consistent regularised risk minimisation possible in polylogarithmic time on polynomially many processors. A similar question, for the case of learning half spaces, has been called a fundamental open problem by Long and Servedio [21] who gave an algorithms which runs on polynomially many processors in time that depends polylogarithmically on the sample size but is inversely proportional to a parameter of the learning problem. While Nick's Class as a notion of efficiency has been criticised [17], it is the only notion of efficiency that forms a proper complexity class in the sense of Blum [4]. To overcome the weakness of using only this notion, Kruskal et al. [17] suggested to consider also the inefficiency of simulating the parallel algorithm on a single processing unit. We discuss the inefficiency and the speed-up in Appendix A.2.

# 4 Empirical Evaluation

This empirical study compares the Radon machine to state-of-the-art parallel machine learning algorithms from the Spark machine learning library [25], as well as the natural baseline of averaging hypotheses instead of calculating their Radon point (averaging-at-the-end, *Avg*). We use base learning algorithms from WEKA [44] and scikit-learn [29]. We compare the Radon machine to the base learning algorithms on moderately sized datasets, due to scalability limitations of the base learners, and reserve larger datasets for the comparison with parallel learners. The experiments are executed on a Spark cluster (5 worker nodes, 25 processors per node)[2]. All results are obtained using 10-fold cross validation. We apply the Radon machine with parameter $h = 1$ and the maximal parameter $h$ such that each instance of the base learning algorithm is executed on a subset of size at least 100 (denoted $h = max$). Averaging-at-the-end executes the base learning algorithm on the same number of subsets $r^h$ as the Radon machine with that parameter and is denoted in the Figures by stating the parameter $h$ as for the Radon machine. All other parameters of the learning algorithms are optimised on an independent split of the datasets. See Appendix B for additional details.

**What is the speed-up of our scheme in practice?** In Figure 1(a), we compare the Radon machine to its base learners on moderately sized datasets (details on the datasets are provided in Appendix B).

https://bitbucket.org/Michael_Kamp/radonmachine.

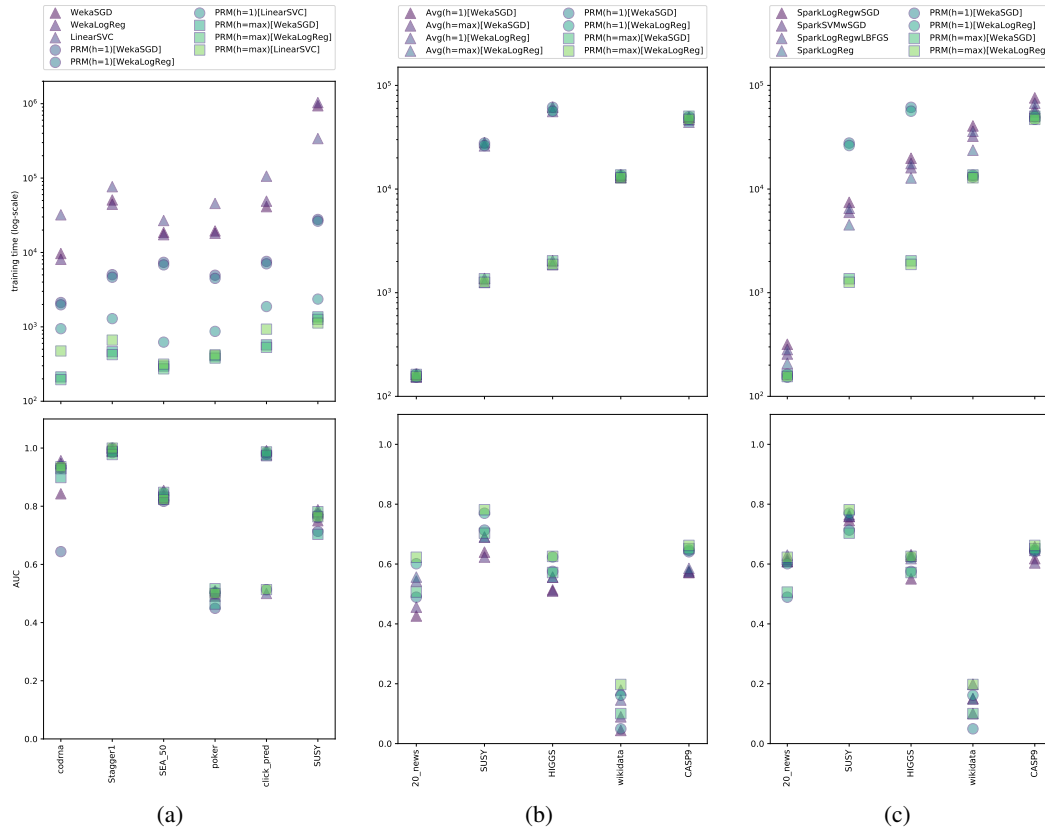

Figure 1: (a) Runtime (log-scale) and AUC of base learners and their parallelisation using the Radon machine (PRM) for 6 datasets with $N \in [488\,565, 5\,000\,000]$, $d \in [3, 18]$. Each point represents the average runtime (upper part) and AUC (lower part) over 10 folds of a learner—or its parallelisation—on one datasets. (b) Runtime and AUC of the Radon machine compared to the averaging-at-the-end baseline (Avg) on 5 datasets with $N \in [5\,000\,000, 32\,000\,000]$, $d \in [18, 2\,331]$. (c) Runtime and AUC of several Spark machine learning library algorithms and the Radon machine using base learners that are comparable to the Spark algorithms on the same datasets as in Figure 1(b).

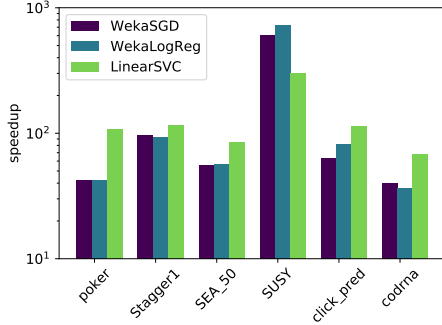

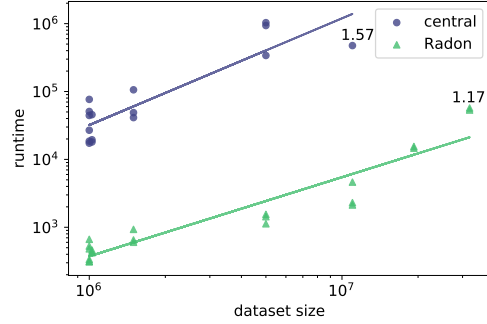

Figure 2: Speed-up (log-scale) of the Radon machine over its base learners per dataset from the same experiment as in Figure 1(a).

Figure 3: Dependence of the runtime on the dataset size for of the Radon machine compared to its base learners.

There, the Radon machine is between 80 and around 700-times faster than the base learner using 150 processors. The speed-up is detailed in Figure 2. On the SUSY dataset (with $5\,000\,000$ instances and $18$ features), the Radon machine on 150 processors with $h = 3$ is 721 times faster than its base learning algorithms. At the same time, their predictive performances, measured by the area under the ROC curve (AUC) on an independent test dataset, are comparable.

**How does the scheme compare to averaging-at-the-end?** In Figure 1(b) we compare the runtime and AUC of the parallelisation scheme against the averaging-at-the-end baseline (*Avg*). In terms of the AUC, the Radon machine outperforms the averaging-at-the-end baseline on all datasets by at least $10\%$. The runtimes can hardly be distinguished in that figure. A small difference can however be noted in Figure 4 which is discussed in more details in the next paragraph. Since averaging is less computationally expensive than calculating the Radon point, the runtimes of the averaging-at-the-end baselines are slightly lower than the ones of the Radon machine. However, compared to the computational complexity of executing the base learner, this advantage becomes negligible.

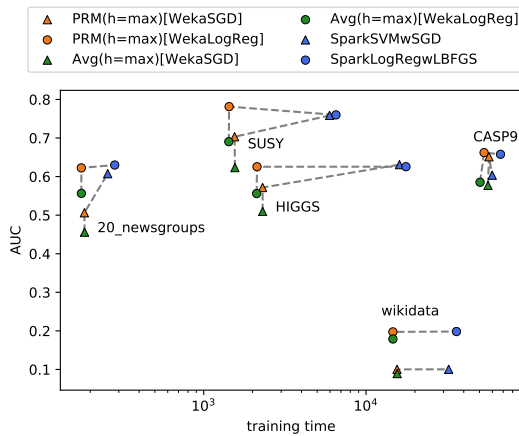

Figure 4: Representation of the results in Figure 1(b) and 1(c) in terms of the trade-off between runtime and AUC for the Radon machine (PRM) and averaging-at-the-end (Avg), both with parameter $h = max$, and parallel machine learning algorithms in Spark. The dashed lines connect the Radon machine to averaging-at-the-end with the same base learning algorithm and a comparable Spark machine learning algorithm.

**How does our scheme compare to state-of-the-art Spark machine learning algorithms?** We compare the Radon machine to various Spark machine learning algorithms on 5 large datasets. The results in Figure 1(c) indicate that the proposed parallelisation scheme with $h = max$ has a substantially smaller runtime than the Spark algorithms on all datasets. On the SUSY and HIGGS dataset, the Radon machine is one order of magnitude faster than the Spark implementations—here the comparatively small number of features allows for a high level of parallelism. On the CASP9 dataset, the Radon machine is $15\%$ faster than the fastest Spark algorithm. The performance in terms of AUC of the Radon machine is similar to the Spark algorithms. In particular, when using WekaLogReg with $h = max$, the Radon machine outperforms the Spark algorithms in terms of AUC and runtime on the datasets SUSY, wikidata, and CASP9. Details are given in the Appendix B. A summarizing comparison of the parallel approaches in terms of their trade-off between runtime and predictive performance is depicted in Figure 4. Here, results are shown for the Radon machine and averaging-at-the-end with parameter $h = max$ and for the two Spark algorithms most similar to the base

learning algorithms. Note that it is unclear what caused the consistently weak performance of all algorithms on wikidata. Nonetheless, the results show that on all datasets the Radon machine has comparable predictive performance to the Spark algorithms and substantially higher predictive performance than averaging-at-the-end. At the same time, the Radon machine has a runtime comparable to averaging-at-the-end on all datasets and both are substantially faster than the Spark algorithms.

**How does the runtime depend on the dataset size in a real-world system?** The runtime of the Radon machine can be distinguished into its learning phase and its aggregation phase. While the learning phase fully benefits from parallelisation, this comes at the cost of additional runtime for the aggregation phase. The time for aggregating the hypotheses does not depend on the number of instances in the dataset but for a fixed parameter $h$ it only depends on the dimension of the hypothesis space and that parameter. In Figure 3 we compare the runtimes of all base learning algorithms per dataset size to the Radon machines. Results indicate that, while the runtimes of the base learning algorithms depends on the dataset size with an average exponent of $1.57$, the runtime of the Radon machine depends on the dataset size with an exponent of only $1.17$.

**How generally applicable is the scheme?** As an indication of the general applicability in practice, we also consider regression and multi-class classification. For regression, we apply the scheme to the Scikit-learn implementation of regularised least squares regression [29]. On the dataset *YearPredictionMSD*, regularised least squares regression achieves an RMSE of $12.57$, whereas the Radon machine achieved an RMSE of $13.64$. At the same time, the Radon machine is 197-times faster. We also compare the Radon machine on a multi-class prediction problem using conditional maximum entropy models. For multi-class classification, we use the implementation described in Mcdonald et al. [23], who propose to use averaging-at-the-end for distributed training. We compare the Radon machine to averaging-at-the-end with conditional maximum entropy models on two large multi-class datasets (*drift* and *spoken-arabic-digit*). On average, our scheme performs better with only slightly longer runtime. The minimal difference in runtime can be explained—similar to the results in Figure 1(b)—by the smaller complexity of calculating the average instead of the Radon point.

## 5   Discussion and Future Work

In the experiments we considered datasets where the number of dimensions is much smaller than the number of instances. **What about high-dimensional models?** The basic version of the parallelisation scheme presented in this paper cannot directly be applied to cases in which the size of the dataset is not at least a multiple of the Radon number of the hypothesis space. For various types of data such as text, this might cause concerns. However, random projections [15] or low-rank approximations [2, 28] can alleviate this problem and are already frequently employed in machine learning. An alternative might be to combine our parallelisation scheme with block coordinate descent [37]. In this case, the scheme can be applied iteratively to subsets of the features.

In the experiments we considered only linear models. **What about non-linear models?** Learning non-linear models causes similar problems to learning high-dimensional ones. In non-parametric methods like kernel methods, for instance, the dimensionality of the optimisation problem is equal to the number of instances, thus prohibiting the application of our parallelisation scheme. However, similar low-rank approximation techniques as described above can be applied with non-linear kernels [11]. Furthermore, methods for speeding up the learning process for non-linear models explicitly approximate an embedding in which then a linear model can be learned [31]. Using explicitly constructed feature spaces, Radon machines can directly be applied to non-linear models.

We have theoretically analysed our parallelisation scheme for the case that there are enough processing units available to find each weak hypothesis on a separate processing units. **What if there are not $r^h$, but only $c < r^h$ processing units?** The parallelisation scheme can quite naturally be "de-parallelised" and partially executed in sequence. For the runtime this implies an additional factor of $\max\{1, r^h/c\}$. Thus, the Radon machine can be applied with any number of processing units.

The scheme improves $\Delta$ doubly exponentially in its parameter $h$ but for that it requires the weak hypotheses to already achieve $\delta \leq 1/2r$. **Is the scheme only applicable in high-confidence domains?** Many application scenarios require high-confidence error bounds, e.g., in the medical domain [27] or in intrusion detection [36]. In practice our scheme achieves similar predictive quality much faster than its base learner.

Besides runtime, communication plays an essential role in parallel learning. **What is the communication complexity of the scheme?** As for all aggregation at the end strategies, the overall amount of communication is low compared to periodically communicating schemes. For the parallel aggregation of hypotheses, the scheme requires $\mathcal{O}(r^{h+1})$ messages (which can be sent in parallel) each containing a single hypothesis of size $\mathcal{O}(r)$. Our scheme is ideally suited for inherently distributed data and might even mitigate privacy concerns.

In a lot of applications data is available in the form of potentially infinite data streams. **Can the scheme be applied to distributed data streams?** For each data stream, a hypotheses could be maintained using an online learning algorithm and periodically aggregated using the Radon machine, similar to the federated learning approach proposed by McMahan et al. [24].

In this paper, we investigated the parallelisation of machine learning algorithms. **Is the Radon machine more generally applicable?** The parallelisation scheme could be applied to more general randomized convex optimization algorithms with unknown and random target functions. We will investigate its applicability for learning in non-Euclidean, abstract convexity spaces.

## 6  Conclusion and Related Work

In this paper we provided a step towards answering an open problem: *Is parallel machine learning possible in polylogarithmic time using a polynomial number of processors only?* This question has been posed for half-spaces by Long and Servedio [21] and called "a fundamental open problem about the abilities and limitations of efficient parallel learning algorithms". It relates machine learning to Nick's Class of parallelisable decision problems and its variants [13]. Early theoretical treatments of parallel learning with respect to NC considered *probably approximately correct* (PAC) [5, 39] concept learning. Vitter and Lin [42] introduced the notion of *NC-learnable* for concept classes for which there is an algorithm that outputs a probably approximately correct hypothesis in polylogarithmic time using a polynomial number of processors. In this setting, they proved positive and negative learnability results for a number of concept classes that were previously known to be PAC-learnable in polynomial time. More recently, the special case of learning half spaces in parallel was considered by Long and Servedio [21] who gave an algorithm for this case that runs on polynomially many processors in time that depends polylogarithmically on the size of the instances but is inversely proportional to a parameter of the learning problem. Our paper complements these theoretical treatments of parallel machine learning and provides a provably effective parallelisation scheme for a broad class of regularised risk minimisation algorithms.

Some parallelisation schemes also train learning algorithms on small chunks of data and average the found hypotheses. While this approach has advantages [12, 32], current error bounds do not allow a derivation of polylogarithmic runtime [20, 35, 45] and it has been doubted to have any benefit over learning on a single chunk [34]. Another popular class of parallel learning algorithms is based on stochastic gradient descent, targeting expected risk minimisation directly [34, and references therein]. The best so far known algorithm in this class [34] is the distributed mini-batch algorithm [10]. This algorithm still runs for a number of rounds inversely proportional to the desired optimisation error, hence not in polylogarithmic time. A more traditional approach is to minimise the *empirical risk*, i.e., an empirical sample-based approximation of the expected risk, using any, deterministic or randomised, optimisation algorithm. This approach relies on generalisation guarantees relating the expected and empirical risk minimisation as well as a guarantee on the optimisation error introduced by the optimisation algorithm. The approach is readily parallelisable by employing available parallel optimisation algorithms [e.g., 6]. It is worth noting that these algorithms solve a harder than necessary optimisation problem and often come with prohibitively high communication cost in distributed settings [34]. Recent results improve over these [22] but cannot achieve polylogarithmic time as the number of iterations depends linearly on the number of processors.

Apart from its theoretical advantages, the Radon machine also has several practical benefits. In particular, it is a black-box parallelisation scheme in the sense that it is applicable to a wide range of machine learning algorithms and it does not depend on the implementation of these algorithms. It speeds up learning while achieving a similar hypothesis quality as the base learner. Our empirical evaluation indicates that in practice the Radon machine achieves either a substantial speed-up or a higher predictive performance than other parallel machine learning algorithms.

# 7 Proof of Proposition 2 and Theorem 3

In order to prove Proposition 2 and consecutively Theorem 3, we first investigate some properties of Radon points and convex functions. We proof these properties for the more general case of quasi-convex functions. Since every convex function is also quasi-convex, the results hold for convex functions as well. A quasi-convex function is defined as follows.

**Definition 5.** *A function $\mathcal{Q} : \mathcal{F} \to \mathbb{R}$ is called* quasi-convex *if all its sublevel sets are convex, i.e.,*

$$\forall \theta \in \mathbb{R} : \{f \in \mathcal{F} \mid \mathcal{Q}(f) < \theta\} \text{ is convex.}$$

First we give a different characterisation of quasi-convex functions.

**Proposition 6.** *A function $\mathcal{Q} : \mathcal{F} \to \mathbb{R}$ is quasi-convex if and only if for all $S \subseteq \mathcal{F}$ and all $s' \in \langle S \rangle$ there exists an $s \in S$ with $\mathcal{Q}(s) \geq \mathcal{Q}(s')$.*

*Proof.*

- ($\Rightarrow$) Suppose this direction does not hold. Then there is a quasi-convex function $\mathcal{Q}$, a set $S \subseteq \mathcal{F}$, and an $s' \in \langle S \rangle$ such that for all $s \in S$ it holds that $\mathcal{Q}(s) < \mathcal{Q}(s')$ (therefore $s' \notin S$). Let $C = \{c \in \mathcal{F} \mid \mathcal{Q}(c) < \mathcal{Q}(s')\}$. As $S \subseteq C = \langle C \rangle$ we also have that $\langle S \rangle \subseteq \langle C \rangle$ which contradicts $\langle S \rangle \ni s' \notin C$.

- ($\Leftarrow$) Suppose this direction does not hold. Then there exists an $\varepsilon$ such that $S = \{s \in \mathcal{F} \mid \mathcal{Q}(s) < \varepsilon\}$ is not convex and therefore there is an $s' \in \langle S \rangle \setminus S$. By assumption $\exists s \in S : \mathcal{Q}(s) \geq \mathcal{Q}(s')$. Hence $\mathcal{Q}(s') < \varepsilon$ and we have a contradiction since this would imply $s' \in S$.

$\square$

The next proposition concerns the value of any convex function at a Radon point.

**Proposition 7.** *For every set $S$ with Radon point $\mathfrak{r}$ and every quasi-convex function $\mathcal{Q}$ it holds that $|\{s \in S \mid \mathcal{Q}(s) \geq \mathcal{Q}(\mathfrak{r})\}| \geq 2$.*

*Proof.* We show a slightly stronger result: Take any family of pairwise disjoint sets $A_i$ with $\bigcap_i \langle A_i \rangle \neq \emptyset$ and $\mathfrak{r} \in \bigcap_i \langle A_i \rangle$. From proposition 6 follows directly the existence of an $a_i \in A_i$ such that $\mathcal{Q}(a_i) \geq \mathcal{Q}(\mathfrak{r})$. The desired result follows then from $a_i \neq a_j \Leftarrow i \neq j$. $\square$

We are now ready to proof Proposition 2 and Theorem 3 (which we re-state here for convenience).

**Theorem 3.** *Given a probability measure $P$ over a hypothesis space $\mathcal{F}$ with finite Radon number $r$, let $F$ denote a random variable with distribution $P$. Denote by $\mathfrak{r}_1$ the random variable obtained by computing the Radon point of $r$ random points drawn iid according to $P$ and by $P_1$ its distribution. For any $h \in \mathbb{N}$, let $\mathfrak{r}_h$ denote the Radon point of $r$ random points drawn iid from $P_{h-1}$ and by $P_h$ its distribution. Then for any convex function $\mathcal{Q} : \mathcal{F} \to \mathbb{R}$ and all $\varepsilon \in \mathbb{R}$ it holds that*

$$P(\mathcal{Q}(\mathfrak{r}_h) > \varepsilon) \leq (rP(\mathcal{Q}(F) > \varepsilon))^{2^h} .$$

*Proof of Proposition 2 and Theorem 3.* By proposition 7, for any Radon point $\mathfrak{r}$ of a set $S$ there must be two points $a, b \in S$ with $\mathcal{Q}(a), \mathcal{Q}(b) \geq \mathcal{Q}(\mathfrak{r})$. Henceforth, the probability of $\mathcal{Q}(\mathfrak{r}) > \varepsilon$ is less than or equal to the probability of the pair $a, b$ having $\mathcal{Q}(a), \mathcal{Q}(b) > \varepsilon$. Proposition 2 follows by an application of the union bound on all pairs from $S$. Repeated application of the proposition proves Theorem 3. $\square$

### Acknowledgements

Part of this work was conducted while Mario Boley, Olana Missura, and Thomas Gärtner were at the University of Bonn and partially funded by the German Science Foundation (DFG, under ref. GA 1615/1-1 and GA 1615/2-1). The authors would like to thank Dino Oglic, Graham Hutton, Roderick MacKenzie, and Stefan Wrobel for valuable discussions and comments.

## Footnotes

[1] We derive $\alpha_\varepsilon, \beta_\varepsilon$ for hypothesis spaces with finite VC [41] and Rademacher [3] complexity in App. C.2.

[2]The source code implementation in Spark can be found in the bitbucket repository

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
