[Supplementary Material]

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

# A  Theory

## A.1  Proof of Theorem 4

In the following, Theorem 4 is proven we which re-state here for convenience.

**Theorem 4.** *The Radon machine with a consistent and efficient regularised risk minimisation algorithm on a hypothesis space with finite Radon number has polylogarithmic runtime on quasi-polynomially many processing units if the Radon number can be upper bounded by a function poly-logarithmic in the sample complexity of the efficient regularised risk minimisation algorithm.y*

*Proof.* We assume the base learning algorithm $\mathcal{L}$ to be a consistent and efficient regularised risk minimisation algorithm on a hypothesis space with finite Radon number. Let $r \in \mathbb{N}$ be the Radon number of the hypothesis space and

$$N_0^{\mathcal{L}}(\varepsilon, \delta) = \left( \alpha_\varepsilon + \beta_\varepsilon \operatorname{ld} \frac{1}{\delta} \right)^k$$

be its sample complexity with $\alpha_\varepsilon, \beta_\varepsilon \geq 0$. In the following, we want to compare the runtime of the Radon machine for achieving an $(\varepsilon, \Delta)$-guarantee to the runtime of the application of the base learning algorithm for achieving the same $(\varepsilon, \Delta)$-guarantee.

To achieve an $(\varepsilon, \Delta)$-guarantee, the Radon machine with parameter $h \in \mathbb{N}$ requires $N = nr^h$ examples (i.e., with $r^h$ processing units), where $n$ denotes the size of the data subset available to each parallel instance of the base learning algorithm. Since $\Delta = (r\delta)^{2^h}$, each base learning algorithm needs to achieve an $(\varepsilon, \delta)$-guarantee and thus requires at least

$$n = \left\lceil N_0^{\mathcal{L}}(\varepsilon, \delta) \right\rceil \leq \left( \alpha_\varepsilon + \beta_\varepsilon \operatorname{ld} \frac{1}{\delta} \right)^k + 1 \tag{8}$$

examples. The application of the base learning algorithm requires at least (cf. Equation 6)

$$M = \left\lceil N_0^{\mathcal{L}}(\varepsilon, \Delta) \right\rceil = \left\lceil \left( \alpha_\varepsilon + 2^h \cdot \beta_\varepsilon \operatorname{ld} \frac{1}{r\delta} \right)^k \right\rceil = \left\lceil \left( \alpha_\varepsilon + 2^h \left( \beta_\varepsilon \operatorname{ld} \frac{1}{\delta} - \beta_\varepsilon \operatorname{ld} r \right) \right)^k \right\rceil . \tag{9}$$

Solving Equation 8 for $\beta_\varepsilon \operatorname{ld} {}^1\!/\!_\delta$ yields

$$\beta_\varepsilon \operatorname{ld} \frac{1}{\delta} \leq (n-1)^{\frac{1}{k}} - \alpha_\varepsilon .$$

By inserting this into Equation 9 we obtain

$$M \geq \left\lceil \left( \alpha_\varepsilon + 2^h \left( (n-1)^{\frac{1}{k}} - \alpha_\varepsilon - \beta_\varepsilon \operatorname{ld} r \right) \right)^k \right\rceil \in \mathcal{O}\left( 2^h \left( n - \operatorname{ld} r \right) \right) . \tag{10}$$

In the following, we show that for the choice of

$$h = \left\lceil \frac{1}{k} \left( \operatorname{ld} M - \operatorname{ld} \operatorname{ld} M \right) \right\rceil , \tag{11}$$

the runtime of the Radon machine is polylogarithmic in $M$, i.e., polylogarithmic in the number of examples the base learning algorithm requires to achieve the same $(\varepsilon, \Delta)$-guarantee. For that, the Radon machine requires quasi-polynomially many processors in $M$. Note that the Radon machine processes $N \geq M$ many samples to achieve that $(\varepsilon, \Delta)$-guarantee, which is more than the base learning algorithm requires by a factor in $\mathcal{O}\left( r^h / 2^{hk} \right)$.

Thus, we need to express the runtime of the Radon machine, that is,

$$T_{\mathcal{R},h}(N) = T_{\mathcal{L}}\left( \frac{N}{r^h} \right) + r^3 \log_r r^h = T_{\mathcal{L}}(n) + r^3 \log_r r^h ,$$

in terms of $M$ instead of $N$. First, we express $n$ in terms of $M$, by solving Equation 10 for $n$ which yields

$$n \leq \left( \left( \alpha_\varepsilon \left( 1 - \frac{1}{2^h} \right) + \beta_\varepsilon \operatorname{ld} r + \frac{1}{2^h} M^{\frac{1}{k}} \right)^k + 1 \right) \in \mathcal{O}\left( \log_2^k r + \frac{1}{2^{hk}} M \right) . \tag{12}$$

Since $\mathcal{L}$ is efficient, $T_{\mathcal{L}}(n) \in \mathcal{O}(n^{\kappa})$ and thus the runtime of the Radon machine in terms of $M$, denoted $T_{\mathcal{R}}^{M}$, is

$$T_{\mathcal{R}}^{M} = T_{\mathcal{L}}(n) + r^3 \log_r r^h \in \mathcal{O}\left(\left(\log_2^k r + \frac{1}{2^{hk}} M\right)^{\kappa} + r^3 \operatorname{ld} r^h\right) \ .$$

Inserting $h$ as in Equation 11 yields

$$\left(\log_2^k r + \frac{1}{2^{hk}} M\right)^{\kappa} + r^3 \operatorname{ld} \frac{M}{\operatorname{ld} M} = \left(\log_2^k r + \frac{M}{2^{k\frac{1}{k} \operatorname{ld} \frac{M}{\operatorname{ld} M}}}\right)^{\kappa} + r^3 \operatorname{ld} \frac{M}{\operatorname{ld} M}$$

$$= \left(\log_2^k r + \frac{M}{\frac{M}{\operatorname{ld} M}}\right)^{\kappa} + r^3 \operatorname{ld} \frac{M}{\operatorname{ld} M}$$

$$= \left(\log_2^k r + \operatorname{ld} M\right)^{\kappa} + r^3 \operatorname{ld} \frac{M}{\operatorname{ld} M} \ .$$

This shows that

$$T_{\mathcal{R}}^{M} \in \mathcal{O}\left(\operatorname{ld}^{\kappa} M + \operatorname{ld}^{k\kappa} r + r^3 \operatorname{ld} M\right) \ .$$

Thus, the runtime of the Radon machine to achieve an $(\varepsilon, \Delta)$-guarantee in terms of $M$ (i.e., the number of samples required by the base learning algorithm to achieve that guarantee) is in $\mathcal{O}\left(\operatorname{ld}^{\kappa} M + \operatorname{ld}^{k\kappa} r + r^3 \operatorname{ld} M\right)$ and therefore polylogarithmic in $M$.

We now determine the number of processing units $c = r^h$ in terms of $M$. For that, observe that $h$ as in Equation 11 can be expressed as

$$h = \left\lceil \frac{1}{k}(\operatorname{ld} M - \operatorname{ld} \operatorname{ld} M) \right\rceil = \left\lceil \frac{1}{k}\left(\operatorname{ld} \frac{M}{\operatorname{ld} M}\right) \right\rceil = \left\lceil \frac{\operatorname{ld} r}{k} \log_r \frac{M}{\operatorname{ld} M} \right\rceil \ .$$

With this the number of processing units is

$$c = r^h \in \mathcal{O}\left(M^{\operatorname{ld} r}\right) \ .$$

$\square$

As mentioned in Section 3, for the Radon machine to achieve an $(\varepsilon, \Delta)$-guarantee each instance of its base learning algorithm has to achieve $\delta \leq 1/2r$. Thus, the sample size with respect to $M$ has to be large enough so that each base learner achieves this minimum $\delta$. Similar to the proof of Theorem 4, we can express this minimum sample size in terms of $M$: The base learning algorithm achieves $\delta \leq 1/2r$ for $M \geq 2^{k\beta_{\varepsilon}(\alpha_{\varepsilon}+1)}$. This can be shown by first observing that Equation 9 implies that for each instance of the base learning algorithm to achieve $\delta \leq 1/2r$ it is required that

$$M \geq \left(\alpha_{\varepsilon} + 2^h \cdot \beta_{\varepsilon} \operatorname{ld} \frac{1}{r^{\frac{1}{2r}}}\right)^k = \left(\alpha_{\varepsilon} + 2^h \beta_{\varepsilon}\right)^k = \left(\alpha_{\varepsilon} + \left(\frac{M}{\operatorname{ld} M}\right)^{\frac{1}{k}} \beta_{\varepsilon}\right)^k \ . \tag{13}$$

This holds for $M \geq 2^{k\beta_{\varepsilon}(\alpha_{\varepsilon}+1)} \geq 2^{k\beta_{\varepsilon}}$, since

$$M \underbrace{\geq}_{\operatorname{ld} M \geq \beta_{\varepsilon}^k (\alpha_{\varepsilon}+1)^k} \frac{M}{\operatorname{ld} M} \beta_{\varepsilon}^k (\alpha_{\varepsilon}+1)^k$$

$$\underbrace{\geq}_{\left(\frac{M}{\operatorname{ld} M}\right)^{\frac{1}{k}} \beta_{\varepsilon} \geq 1} \left(\left(\frac{M}{\operatorname{ld} M}\right)^{\frac{1}{k}} \beta_{\varepsilon}\left(\frac{\alpha_{\varepsilon}}{\left(\frac{M}{\operatorname{ld} M}\right)^{\frac{1}{k}} \beta_{\varepsilon}} + 1\right)\right)^k = \left(\alpha_{\varepsilon} + \left(\frac{M}{\operatorname{ld} M}\right)^{\frac{1}{k}} \beta_{\varepsilon}\right)^k \ .$$

After having proven that the Radon machine has polylogarithmic runtime on quasi-polynomially many processors, in the following section we analyse the speed-up over the base learning algorithm which achieves the same $(\varepsilon, \Delta)$-guarantee.

## A.2 Analysis of the Speed-Up of the Radon machine

In this section, we analyse the speed-up of the Radon machine over the execution of the base learning algorithm when both achieve the same $(\varepsilon, \Delta)$-guarantee, as well as its inefficiency [17] and its data inefficiency, i.e., how much more data the Radon machine requires compared to the base learning algorithm which achieves the same $(\varepsilon, \Delta)$-guarantee. For that, recall that the sample complexity of the base learning algorithm for a given $\varepsilon > 0, 0 < \Delta < 1$ is

$$N_0^{\mathcal{L}}(\varepsilon, \Delta) = \left( \alpha_\varepsilon + \beta_\varepsilon \log_2 \frac{1}{\Delta} \right)^k .$$

We assume that $\alpha_\varepsilon \in \Theta(\varepsilon^{-1})$ and $\beta_\varepsilon \in \Theta(\varepsilon^{-1})$ (see for example Lemma 11 and Lemma 12). Following the notion of Hanneke [14] the sample complexity can then be expressed as

$$N_0^{\mathcal{L}}(\varepsilon, \Delta) \in \Theta \left( \left( \frac{1}{\varepsilon} + \frac{1}{\varepsilon} \operatorname{ld} \frac{1}{\Delta} \right)^k \right) = \Theta \left( \left( \frac{1}{\varepsilon} \operatorname{ld} \frac{1}{\Delta} \right)^k \right) . \tag{14}$$

Similar to Kruskal et al. [17], we assume the base algorithm to have a runtime polynomial in $N$, i.e.,

$$T_{\mathcal{L}} \in \Theta \left( N^\kappa \right) = \Theta \left( \left( \frac{1}{\varepsilon} \operatorname{ld} \frac{1}{\Delta} \right)^{k\kappa} \right) . \tag{15}$$

The Radon machine runs $\mathcal{L}$ in parallel on $c$ processors to obtain $r^h$ weak hypotheses with $(\varepsilon, \delta)$-guarantee. It then combines the obtained solutions $h$ times—level-wise in parallel—calculating the Radon point (which takes time $r^3$). In this paper we assume the number of available processors to be abundant and thus set $c = r^h$. With this, the runtime of the Radon machine is

$$T_{\mathcal{R}} \in \Theta \left( \left( \frac{1}{\varepsilon} \operatorname{ld} \frac{1}{\delta} \right)^{k\kappa} + hr^3 \right) . \tag{16}$$

We now provide an analysis on the speed-up for $c = r^h$ and arbitrary $h \in \mathbb{N}$.

**Proposition 8.** *Given a polynomial time consistent regularised risk minimisation algorithm $\mathcal{L}$ using a hypothesis space with finite Radon number $r \in \mathbb{N}$ and runtime as in Equation 15, the Radon machine run with parameter $h \in \mathbb{N}$ on $r^h$ processors. Then, the ratio of the runtime of the base learner over the runtime of the Radon machine, denoted the speed-up [17]*

$$\frac{T_{\mathcal{L}}}{T_{\mathcal{R}}} ,$$

*is in*

$$\Theta \left( \frac{2^{hk\kappa}}{1 + \frac{hr^3}{\left( \frac{1}{\varepsilon} \operatorname{ld} \frac{1}{\delta} \right)^{k\kappa}}} \right) .$$

*Proof.* In order to achieve an $(\varepsilon, \Delta)$-guarantee, the Radon machine runs $r^h$ parallel instances of the the base learning algorithm on $n = \lceil N_0^{\mathcal{L}}(\varepsilon, \delta) \rceil$ examples with $\delta \leq 1/2r$ so that $\Delta = (r\delta)^{2^h}$. To achieve the same $(\varepsilon, \Delta)$-guarantee, the base learning algorithm requires

$$M = \lceil N_0^{\mathcal{L}}(\varepsilon, \Delta) \rceil = \left\lceil \left( 2^h \cdot \frac{1}{\varepsilon} \operatorname{ld} \frac{1}{r\delta} \right)^k \right\rceil \in \Theta \left( \left( 2^h \frac{1}{\varepsilon} \operatorname{ld} \frac{1}{r\delta} \right)^k \right) = \Theta \left( \left( 2^h \frac{1}{\varepsilon} \operatorname{ld} \frac{1}{\delta} \right)^k \right)$$

many examples. The last step follows from the fact that, since $\delta \leq 1/2r$, we have $1/r\delta \geq 2r/r \geq r$ and thus

$$\operatorname{ld} \frac{1}{r\delta} \leq \operatorname{ld} \frac{1}{r\delta} + \operatorname{ld} r = \operatorname{ld} \frac{1}{\delta} \leq 2 \operatorname{ld} \frac{1}{r\delta}$$

$$\Rightarrow \operatorname{ld} \frac{1}{\delta} \in \Theta \left( \operatorname{ld} \frac{1}{r\delta} \right) \Leftrightarrow \operatorname{ld} \frac{1}{r\delta} \in \Theta \left( \operatorname{ld} \frac{1}{\delta} \right) ,$$

To achieve the $(\varepsilon, \Delta)$-guarantee, the base learning algorithm has a runtime of

$$T_{\mathcal{L}} \in \Theta\left(M^{\kappa}\right) = \Theta\left(\left(2^h \frac{1}{\varepsilon} \operatorname{ld} \frac{1}{\delta}\right)^{k\kappa}\right) \quad .$$

Using $T_{\mathcal{R}}$ from Equation 16, we get that

$$\frac{T_{\mathcal{R}}}{T_{\mathcal{L}}} \in \Theta\left(\frac{\left(\frac{1}{\varepsilon} \operatorname{ld} \frac{1}{\delta}\right)^{k\kappa} + hr^3}{\left(2^h \frac{1}{\varepsilon} \operatorname{ld} \frac{1}{\delta}\right)^{k\kappa}}\right) = \Theta\left(\frac{1}{2^{hk\kappa}}\left(1 + \frac{hr^3}{\left(\frac{1}{\varepsilon} \operatorname{ld} \frac{1}{\delta}\right)^{k\kappa}}\right)\right)$$

The speed-up then is

$$\frac{T_{\mathcal{L}}}{T_{\mathcal{R}}} \in \Theta\left(\frac{2^{hk\kappa}}{1 + \frac{hr^3}{\left(\frac{1}{\varepsilon} \operatorname{ld} \frac{1}{\delta}\right)^{k\kappa}}}\right) \quad .$$

$\square$

Note that the runtime of the Radon machine for the case that $1 \le c \le r^h$ is given by

$$T_{\mathcal{R}} \in \Theta\left(\frac{r^h}{c}\left(\left(\frac{1}{\varepsilon} \operatorname{ld} \frac{1}{\delta}\right)^{k\kappa}\right) + r^3 \sum_{i=1}^{h}\left\lceil \frac{r^i}{c}\right\rceil\right) \quad .$$

In this case, the speed-up is lower by a factor of $r^h/c$.

In the following, we analyse the inefficiency [17] of the Radon machine, i.e., the ratio between the total number of operations executed by all processors and the work of the base learning algorithm.

**Proposition 9.** *The Radon machine with a consistent and efficient regularised risk minimisation algorithm on a hypothesis space with finite Radon number has quasi-polynomial inefficiency if the Radon number is upper bounded by a function polylogarithmic in the sample complexity of the efficient regularised risk minimisation algorithm.*

*Proof.* Let $\mathcal{L}$ be a consistent and efficient regularised risk minimisation algorithm on a hypothesis space with finite Radon number $r \in \mathbb{N}$. Since $\mathcal{L}$ is efficient, its runtime $T_{\mathcal{L}}(M)$ is in $\mathcal{O}(M^{\kappa})$. From the proof of Theorem 4 follows that, when choosing $h = \left\lceil \frac{1}{k}\left(\operatorname{ld} M - \operatorname{ld} \operatorname{ld} M\right)\right\rceil$ the Radon machine has a runtime of $T_{\mathcal{R}}(M) \in \mathcal{O}\left(\operatorname{ld}^{\kappa} M + \operatorname{ld}^{k\kappa} r + r^3 \operatorname{ld} M\right)$ using $c \in \mathcal{O}\left(M^{\operatorname{ld} r}\right)$ processing units. The inefficiency of the Radon machine then is

$$\frac{c \cdot T_{\mathcal{R}}(M)}{T_{\mathcal{L}}(M)} \in \mathcal{O}\left(\frac{M^{\operatorname{ld} r}\left(\operatorname{ld}^{\kappa} M + \operatorname{ld}^{k\kappa} r + r^3 \operatorname{ld} M\right)}{M^{\kappa}}\right) \in \mathcal{O}\left(M^{(\operatorname{ld} r)-\kappa} \operatorname{ld}^{\kappa} M\right) = \mathcal{O}\left(M^{\operatorname{ld} r}\right) \quad .$$

Thus, the inefficiency of the Radon machine is quasi-polynomially bounded or, for short, it has quasi-polynomial inefficiency. $\square$

In order to achieve the same $(\varepsilon, \Delta)$-guarantee as the base learning algorithm, the Radon machine requires more data. In the following, we analyse the data inefficiency $N_{\mathcal{R}}(\varepsilon, \Delta)/N_{\mathcal{L}}(\varepsilon, \Delta)$, i.e., the ratio of the data required by the Radon machine over the data required by the base learning algorithm.

**Proposition 10.** *The Radon machine with a consistent and efficient regularised risk minimisation algorithm $\mathcal{L}$ with sample complexity $N_{\mathcal{L}}(\varepsilon, \Delta)$ on a hypothesis space with finite Radon number $r \in \mathbb{N}$ has a data inefficiency in*

$$\Theta\left(\left(\frac{M}{\operatorname{ld} M}\right)^{\frac{\operatorname{ld} r}{k}}\right) \quad ,$$

*where $M = \lceil N_{\mathcal{L}}(\varepsilon, \Delta)\rceil$.*

*Proof.* We assume the sample complexity can be expressed as in Equation 14. For $\Delta = (r\delta)^{2^h}$ we have that

$$
\begin{aligned}
N_{\mathcal{R}}(\varepsilon, \Delta) =& r^h N_{\mathcal{L}}(\varepsilon, \delta) \in \Theta\left(r^h \left(\frac{1}{\varepsilon}\operatorname{ld}\frac{1}{\delta}\right)^k\right) \\
=& \Theta\left(r^h \left(\frac{1}{2^h}\frac{1}{\varepsilon}\operatorname{ld}\frac{1}{\Delta}\right)^k\right) = \Theta\left(\frac{r^h}{2^{hk}}\left(\frac{1}{\varepsilon}\operatorname{ld}\frac{1}{\Delta}\right)^k\right) \\
=& \Theta\left(\frac{r^h}{2^{hk}}N_{\mathcal{L}}(\varepsilon, \Delta)\right) = \Theta\left(\frac{r^h}{2^{hk}}M\right) \quad.
\end{aligned}
$$

Thus, the data inefficiency is in

$$
\Theta\left(\frac{r^h}{2^{hk}}\right) \quad.
$$

Choosing $h = \lceil k^{-1}(\operatorname{ld} M - \operatorname{ld}\operatorname{ld} M)\rceil$ as in the proof of Theorem 4, this is in

$$
\Theta\left(\frac{r^{\frac{1}{k}\operatorname{ld} r}\log_r\frac{M}{\operatorname{ld} M}}{2^{k\frac{1}{k}\operatorname{ld}\frac{M}{\operatorname{ld} M}}}\right) = \Theta\left(\frac{\left(\frac{M}{\operatorname{ld} M}\right)^{\frac{\operatorname{ld} r}{k}}}{\frac{M}{\operatorname{ld} M}}\right) = \Theta\left(\left(\frac{M}{\operatorname{ld} M}\right)^{\frac{\operatorname{ld} r}{k}}\right)
$$

$\square$

## B  Experiments

This section provides additional details on the experiments conducted. All experiments are performed on a Spark cluster with a master node, 5 worker nodes, 25 processors and 64GB of RAM per node. The Radon machine is applied with parameter $h = 1$ and with the maximal $h$ for a given dataset: Recall, that the number of iterations $h$ is limited by the dataset size (i.e., number of instances) and the Radon number of the hypothesis space, since the dataset is partitioned into $r^h$ parts of size $n$. Thus, given a data set of size $N$, the maximal $h$ is given by

$$
h_{\max} = \left\lfloor \log_r \frac{N}{n_{\min}} \right\rfloor \quad,
$$

where $n_{\min}$ denotes the minimum size of the local subset of data that each instance of the base learner is executed on. The experiments have been carried out with $n_{\min} = 100$. As discussed in Section 5, if $r^h$ is larger than the actual number of processing units, some instances of the base learner are executed sequentially.

As base learning algorithms we use the WEKA [44] implementation of Stochastic Gradient Descent (*WekaSGD*), and Logistic Regression (*WekaLogReg*), as well as a the Scikit-learn [29] implementation of the linear support vector machine (*LinearSVM*) with pyspark. The paralellisation of a base learner using the Radon machine is denoted *PRM(h=?)[<base learner>]*.

We compare the Radon machine to the natural baseline of aggregating hypotheses by calculating their average, denoted averaging-at-the-end (*Avg(h=?)[<base learner>]*). Given a parameter $h \in \mathbb{N}$, averaging-at-the-end executes the base learning algorithm on $r^h$ subsets of the data, i.e., on the same number of subsets as the Radon machine. Accordingly, the runtime for obtaining the set of hypotheses is similar, but the time for aggregating the models is shorter, since averaging is less computationally expensive than calculating the Radon point.

We also compare the Radon machine to parallel machine learning algorithms from the Spark machine learning library (MLlib): SparkMLLibLogisticRegressionWithLBFGS (*SparkLogRegwLBFGS*), SparkMLLibLogisticRegressionWithSGD (*SparkLogRegwSGD*), SparkMLLibSVMWithSGD (*SparkSVMwSGD*), and SparkMLLogisticRegression (*SparkLogReg*).

The properties of the datasets used in the empirical evaluation are presented in Table 1. Datasets have been acquired from OpenML [40], the UCI machine learning repository [19], and Big Data competition of the ECDBL'14 workshop[3]. Experiments on moderately sized datasets—on which we compare the Radon machine to the base learning algorithms executed on the entire dataset are conducted

| Name | Instances | Dimensions | Output |
|---|---|---|---|
| click_prediction | 1 496 391 | 11 | $\mathcal{Y} = \{-1, 1\}$ |
| poker | 1 025 010 | 10 | $\mathcal{Y} = \{-1, 1\}$ |
| SUSY | 5 000 000 | 18 | $\mathcal{Y} = \{-1, 1\}$ |
| Stagger1 | 1 000 000 | 9 | $\mathcal{Y} = \{-1, 1\}$ |
| HIGGS | 11 000 000 | 28 | $\mathcal{Y} = \{-1, 1\}$ |
| SEA_50 | 1 000 000 | 3 | $\mathcal{Y} = \{-1, 1\}$ |
| codrna | 488 565 | 8 | $\mathcal{Y} = \{-1, 1\}$ |
| CASP9 | 31 993 555 | 631 | $\mathcal{Y} = \{-1, 1\}$ |
| wikidata | 19 254 100 | 2331 | $\mathcal{Y} = \{-1, 1\}$ |
| 20_newsgroups | 399 940 | 1002 | $\mathcal{Y} = \{-1, 1\}$ |
| YearPredictionMSD | 515 345 | 90 | $\mathcal{Y} \subseteq \mathbb{R}$ |
| drift | 13 991 | 90 | $\mathcal{Y} = \{1, \ldots, 89\}$ |
| spoken-arabic-digit | 263 256 | 15 | $\mathcal{Y} = \{1, \ldots, 10\}$ |

Table 1: Description of the datasets used in our experiments.

on the datasets click_prediction, poker, SUSY, Stagger1, SEA_50, and codrna. The comparison of Radon machine and Spark MLlib learners is executed on the datasets CASP9, HIGGS, wikidata, 20_newsgroups, and SUSY. The regression experiment is conducted using the YearPredictionMSD dataset, multiclass-prediction experiments using the drift, and spoken-arabic-digit datasets.

In the following, we provide more details on the experiments presented in Figures 1(a), 1(b), and 1(c) in Section 4. In particular, we analyse the trade-off between training time and AUC per dataset.

Figure 5 shows the trade-off between training time and AUC for base learning algorithms and their parallelisation using the Radon machine. It confirms that the training time for the Radon machine is orders of magnitude smaller than the base learning algorithms on all datasets. Moreover, the training time is substantially smaller for the Radon machine with maximal height ($h = max$), compared to the parameter $h = 1$. In terms of AUC, the performance of the parallelisation is comparable to the base learner for WekaLogReg and LinearSVC on all datasets. For the base learner WekaSGD, the predictive performance of the parallelisation with the Radon machine is comparable on all datasets but codrna. There, the Radon machine with parameter $h = 1$ has substantially lower AUC, while the parallelisation with $h = max$ has substantially higher AUC than the base learning algorithm executed on the entire dataset.

The comparison of the Radon machine to the averaging-at-the-end baseline in Figure 6 confirms the findings of Section 4, i.e., the Radon machine achieves a substantially higher AUC with only slightly higher runtime. Comparing the Radon machine to the Spark MLlib learning algorithms in Figure 7 indicates that the Radon machine is always favourable in terms of training time. However, in terms of AUC the results are mixed: For the base learner WekaLogReg, its parallelisation is always among the best in terms of AUC. The parallelisation of WekaSGD, however, has worse performance than the Spark learners on 2 out of 5 datasets. It also confirms that for the datasets SUSY and HIGGS, the runtime of the Radon machine with $h = 1$ is substantially larger than for $h = max$. Thus, for the best performance in terms of runtime and AUC, the height should be maximal.

In order to investigate the results depicted in Figure 7 more closely, we provide the training times and AUCs in detail in Table 2. As mentioned above, the Radon machine using WekaLogReg as base learner has better runtime than all Spark algorithms. At the same time, this version of the Radon machine outperforms the Spark algorithms in terms of AUC on all datasets but 20_newsgroups— there it is 2.2% worse than the best Spark algorithm. In particular, on the largest dataset in the

Figure 5: AUC vs. training time for base learning algorithms and their parallelisation with the Radon machine per dataset from the same experiment as in Figure 1(a).

experiments—the CASP9 dataset with 32 million instances and 631 features—the Radon machine is 15% faster and 2.6% better in terms of AUC than the best Spark algorithm.

Note that for HIGGS and SUSY, the Radon machine with $h = 1$ is an order of magnitude slower than with $h = max$ as well as the Spark algorithms. This follows from the low degree of parallelisation, since for $h = 1$ only 20 (for SUSY), respectively 30 (for HIGGS) hypotheses have to be generated. Thus, only 20, or 30 of the 150 available processors are used in parallel. At the same time, the amount of data each processor has to process is orders of magnitude larger than for $h = max$.

For the above experiments we assume that the data is already distributed over the nodes in the cluster so that it can directly be processed by the Radon machine. When loading data in Spark, this data is distributed over the worker nodes in subsetsbut not necessarily in $r^h$ subsets. In Spark, distributed

| Dataset | Runtime | | | | | | | |
|---|---|---|---|---|---|---|---|---|
| | SparkLogReg wSGD | SparkSVM wSGD | PRM(h=1) [WekaSGD] | PRM(h=max) [WekaSGD] | SparkLogReg wLBFGS | SparkLogReg | PRM(h=1) [WekaLogReg] | PRM(h=max) [WekaLogReg] |
| 20_newsgroups | 317.7 | 256.2 | 163.4 | 162.5 | 282.9 | 208.5 | **152.8** | 155.4 |
| SUSY | 7 439.5 | 5 961.8 | 27 781.6 | 1 363.7 | 6 526.3 | 4 516.8 | 26 299.6 | **1 259.7** |
| HIGGS | 19 815.1 | 16 071.9 | 61 429.5 | 2 029.7 | 17 617.4 | 12 783.6 | 56 394.2 | **1 876.2** |
| wikidata | 40 645.8 | 32 288.5 | 13 575.7 | 13 677.3 | 36 060.1 | 23 702.0 | 13 039.5 | **12 845.5** |
| CASP9 | 75 782.4 | 59 864.7 | 49 711.5 | 50 430.6 | 67 367.3 | 55 523.5 | 47 085.1 | **47 070.1** |
| | AUC | | | | | | | |
| 20_newsgroups | 0.6098 | 0.6075 | 0.4893 | 0.5063 | **0.63** | 0.6165 | 0.601 | 0.6226 |
| SUSY | 0.7454 | 0.7585 | 0.7134 | 0.7033 | 0.76 | 0.7652 | 0.7697 | **0.7814** |
| HIGGS | 0.5506 | **0.631** | 0.5753 | 0.5717 | 0.6257 | 0.6181 | 0.6237 | 0.6256 |
| wikidata | 0.1505 | 0.1004 | 0.0494 | 0.1002 | 0.1983 | 0.1489 | 0.1615 | **0.1974** |
| CASP9 | 0.6181 | 0.6037 | 0.641 | 0.6514 | 0.6579 | 0.6454 | 0.6464 | **0.6622** |

Table 2: Runtime and AUC of Spark machine learning library algorithms and the Radon machine using WekaSGD and WekaLogReg as base learning algorithms. The results, reported for each dataset, are the average over all folds in a 10-fold cross-validation. These results correspond to the ones presented in Figure 1(c) in Section 4.

Figure 6: AUC vs. training time for the parallelisation of base learning algorithms using the averaging-at-the-end baseline (*Avg*) and the Radon machine per dataset from the same experiment as in Figure 1(b).

data is organised in partitions, where each partition corresponds to the subset of data available to one instance of the base learning algorithm. In order to apply the Radon machine to a dataset within the Spark framework, the data needs to re-distributed and partitioned into $r^h$ partitions which is achieved by a method called repartition. In the experiments in Section 4, we assume that the data is already partitioned to make a fair comparison to the Spark learning algorithms which do not require repartitioning. Figure 8(a) illustrates the time required for repartitioning a dataset in contrast to the runtime of the Radon machine. Repartitioning in Spark always includes a complete shuffling of the data, requiring communication to redistribute the dataset. This is rather inefficient in our context. Nonetheless, the time required for repartitioning is small compared to the overall runtime— in the worst case it takes $14\%$ of the runtime of the Radon machine. Still, taking into account the time for repartitioning the data shrinks the runtime advantage of the proposed scheme over the Spark algorithms. Figure 8(b) shows the runtimes of the Spark algorithms compared to the Radon machine—similar to Figure 1(c) in Section 4—but with the time required for repartitioning the data added to the runtime of the *Radon machines*. The Radon machine with $h = max$ remains superior to the Spark algorithms in terms of runtime.

# C   Practical Aspects

## C.1   Radon Point Construction

In the following, a simple construction is given for a system of linear equations with which a Radon point of a set can be determined. In his main theorem, Radon [30] gives the following construction of a Radon point for a set $S = \{s_1, ..., s_r\} \subseteq \mathbb{R}^d$. Find a non-zero solution $\lambda \in \mathbb{R}^{|S|}$ for the

Figure 7: AUC vs. training time for Spark learners and parallelisations of comparable base learning algorithms with the Radon machine per dataset from the same experiment as in Figure 1(c).

following linear equations.

$$\sum_{i=1}^{r} \lambda_i s_i = (0, \ldots, 0) \ , \ \sum_{i=1}^{r} \lambda_i = 0$$

Such a solution exists, since $|S| > d+1$ implies that $S$ is linearly dependent. Then, let $I, J$ be index sets such that for all $i \in I : \lambda_i \geq 0$ and for all $j \in J : \lambda_j < 0$. Then a Radon point is defined by

$$\mathfrak{r}(\lambda) = \sum_{i \in I} \frac{\lambda_i}{\Lambda} s_i = \sum_{j \in J} \frac{\lambda_j}{\Lambda} s_j \ ,$$

where $\Lambda = \sum_{i \in I} \lambda_i = -\sum_{j \in J} \lambda_j$. Any solution to this linear system of equations is a Radon point. The equation system can be solved in time $r^3$. By setting the first element of $\lambda$ to one, we obtain a unique solution of the system of linear equations. Using this solution $\lambda$, we define the Radon point of a set $S$ as $\mathfrak{r}(S) = \mathfrak{r}(\lambda)$ in order to resolve ambiguity.

### C.2 Consistency Results for Empirical Risk Minimisation

In this section we provide some technical results on the consistency of empirical risk minimisation algorithms.

**Lemma 11.** *For consistent empirical risk minimisers with a hypothesis class of finite Vapnik-Chervonenkis (VC) dimension the sample size required to achieve an $(\varepsilon, \Delta)$-guarantee is given by $N(\Delta) = (\alpha_\varepsilon + \beta_\varepsilon \log_2 {}^1/\Delta)^k$ with $\alpha_\varepsilon = 4 \ln 2^1/\varepsilon^2, \beta_\varepsilon = {}^4/\varepsilon^2 \log_2 e$ and $k = 2$.*

*Proof.* For consistent empirical risk minimisers with finite VC-dimension, the confidence $1 - \Delta$ for a given $N$ and $\varepsilon$ is $\Delta = 2\mathcal{N}(\mathcal{F}, N) \exp(-N\varepsilon^2/4)$ [43], where the shattering coefficient $\mathcal{N}(\mathcal{F}, N)$

Figure 8: (a) Runtime of the Radon machine together with the time required for repartitioning the data to fit the parallelisation scheme. (b) Runtime and AUC of several Spark machine learning library algorithms and the Radon machine including the time required for repartitioning the data before training.

is a polynomial in $N$ for finite VC-dimension. Solving for $N$ yields that the algorithm run with

$$N \geq \frac{1}{\varepsilon^2}\left(\ln 2 + 4\frac{1}{\log_2(e)}\log_2\frac{1}{\delta}\right)$$

achieves a confidence larger or equal to the desired $1 - \Delta$. $\qquad\square$

**Lemma 12.** *For consistent empirical risk minimisers with a hypothesis class of finite Rademacher complexity the sample size required to achieve an $(\varepsilon, \Delta)$-guarantee is given by $N(\Delta) = (\alpha_\varepsilon + \beta_\varepsilon \log_2 1/\Delta)^k$ with $\alpha_\varepsilon = 0, \beta_\varepsilon = 1/2(\varepsilon + 2\rho)^2$ and $k = 1$, where $\rho$ denotes the Rademacher complexity.*

*Proof.* For consistent empirical risk minimisers with a hypothesis class of finite Rademacher complexity $\rho$, a given $\Delta$ and $N$ the error bound is given by $\varepsilon = 2\rho + \sqrt{\log_2 1/\delta/2N}$ [43]. Solving for $N$ yields the above result. $\qquad\square$

## Footnotes

[1] We derive $\alpha_\varepsilon, \beta_\varepsilon$ for hypothesis spaces with finite VC [41] and Rademacher [3] complexity in App. C.2.

[2]The source code implementation in Spark can be found in the bitbucket repository

[3]Big Data Competition 2014: `http://cruncher.ncl.ac.uk/bdcomp/`