[Reviews · NeurIPS 2017]

Reviewer 1



The main contribution of the paper is a new framework for parallel machine learning algorithms. The idea is to combine base learners more effectively than simply averaging. Specifically, subsets of hypotheses are replaced by their radon point. They prove complexity bounds for their method and also empirically compare their results with parallel algorithms in Spark and using base linear models in Weka. The paper is quite interesting since it proposes essentially a black-box method to combine weak learners together. The approach seems to be an alternative to bagging weak learners together. The theoretical contribution is the analysis of the complexity of the radon machine, where, since the original samples can be broken down into multiple parts, the resulting parallel machine is much more efficient as compared to the base learner that operates on all the original samples.They also show a PAC bound on the radon machine. The theoretical contributions seems to be pretty good in the paper. One major concern was the practical aspect, particularly for high-dimensional data. It is not clear how well the proposed approach performs here. the experiments seem to be for very low dimensional datasets (18 features?) which is perhaps not realistic for modern machine learning problems in most domains. If the authors could emphasize on how high-dimensional could be handled by their method, the paper would be much stronger.

Reviewer 2



The authors proposed an approach for parallelizing learners (which hold certain assumptions) based on Radon points. The contribution of the paper w.r.t. the other similar works is the usage of Radon points instead of averaging (which has been used as baseline approach in the paper). The performance is compared with `no parallelization', `averaging schema', and `Spark-based' algorithms. They have shown experimentally the reduced learning time and similar or higher accuracy to the other algorithms. It seems the theoretical part has already been investigated in another paper:``Parallelizing Randomized Convex Optimization", which could be from the same authors. But this is not cited in the current paper and it needs to be elaborated what the paper's contribution for this part is.

Reviewer 3



The authors have presented a parallelization algorithm for aggregating weak learners based on Radon partitioning. They present theoretical analysis to motivate the algorithm along with empirical results to support the theory. The theoretical analysis is interesting, and the empirical results demonstrate training time and/or AUC improvements over multiple baseline algorithms, on multiple datasets. The authors also preemptively and convincingly address several questions/concerns in the Evaluation and Discussion sections. I recommend acceptance. Specific Notes -------------- - Line 51: "the the" -> "the" - Line 79: Is this the same size math font as elsewhere? It seems small and kind of hard to read. - Figures 1-3: The text really needs to be bigger in the figure images. - Figure 3: This one may be harder to distinguish if printed in black and white. - Lines 318-320: "the Radon machine outperforms the averaging baseline..." To clarify, is this based on an average of paired comparisons? - Line 324: A pet peeve of mine is the use of the word significant in papers when not referring to statistical significance results. - Lines 340-341: Missing some spaces - "machineachieved", and "machineis." - Lines 341-349: The difference in performance with the AverageAtTheEnd method doesn't seem terribly meaningful. Why wasn't this result included somewhere in Figure 1? Can you advocate more for your method here? - Line 377: "Long and servedio [22]" The use of author names here seems inconsistent with citations elsewhere (even immediately following).